:◉: PLOS | ONE

# Mixotrophic cultivation of *Spirulina platensis* in dairy wastewater: Effects on the production of biomass, biochemical composition and antioxidant capacity

**Maria I. B. Pereira**[1☉], **Bruna M. E. Chagas**[1☉¤*], **Roberto Sassi**[2‡], **Guilherme F. Medeiros**[3‡], **Emerson M. Aguiar**[1‡], **Luiz H. F. Borba**[1‡], **Emanuelle P. E. Silva**[1‡], **Júlio C. Andrade Neto**[1‡], **Adriano H. N. Rangel**[1☉]

**1** Agricultural School of Jundiaí, Federal University of Rio Grande do Norte, Natal, RN, Brazil, **2** Department of Systematic Ecology, Federal University of Paraíba, João Pessoa, PB, Brazil, **3** Department of Oceanography and Limnology, Federal University of Rio Grande do Norte, Natal, RN, Brazil

☉ These authors contributed equally to this work.
¤ Current address: Infrastructure Superintendence, Federal University of Rio Grande do Norte, Natal, RN, Brazil.
‡ These authors also contributed equally to this work.
* brunam.emerenciano@gmail.com

**Data Availability Statement:** All relevant data are within the paper and its Supporting Information files.

## Abstract

Mixotrophic cultivation of microalgae provides a very promising alternative for producing carbohydrate-rich biomass to convert into bioethanol and value-added biocompounds, such as vitamins, pigments, proteins, lipids and antioxidant compounds. *Spirulina platensis* may present high yields of biomass and carbohydrates when it is grown under mixotrophic conditions using cheese whey. However, there are no previous studies evaluating the influence of this culture system on the profile of fatty acids or antioxidant compounds of this species, which are extremely important for food and pharmaceutical applications and would add value to the cultivation process. *S. platensis* presented higher specific growth rates, biomass productivity and carbohydrate content under mixotrophic conditions; however, the antioxidant capacity and the protein and lipid content were lower than that of the autotrophic culture. The maximum biomass yield was 2.98 ±0.07 g/L in growth medium with 5.0% whey. The phenolic compound concentration was the same for the biomass obtained under autotrophic and mixotrophic conditions with 2.5% and 5.0% whey. The phenolic compound concentrations showed no significant differences except for that in the growth medium with 10.0% whey, which presented an average value of 22.37±0.14 mg gallic acid/g. Mixotrophic cultivation of *S. platensis* using whey can be considered a viable alternative to reduce the costs of producing *S. platensis* biomass and carbohydrates, shorten cultivation time and produce carbohydrates, as it does not require adding expensive chemical nutrients to the growth medium and also takes advantage of cheese whey, an adverse dairy industry byproduct.

**Funding:** This work is part of the first author's Master Dissertation. The author MIBP received a master's degree fellowship from Coordenação de Aperfeiçoamento de Pessoal de Nível Superior (CAPES).

**Competing interests:** The authors have declared that no competing interests exist.

## Introduction

Microalgae have great biotechnological potential and can be used in several segments of the chemical, food, pharmaceutical and cosmetic industries and for biofuel production [1]. These microorganisms are an abundant source of natural proteins, lipids, carbohydrates, vitamins, pigments and enzymes [2] [3]. Moreover, numerous species are reported to have a significant antioxidant effect attributed to their biocompounds [2]. Antioxidants are compounds that are capable of inhibiting or retarding the oxidation of an oxidizable substrate and have a relevant role in human health, protecting the body against the oxidizing action of free radicals [4]. The antioxidant activity of *S. platensis* has recently been studied, and many in vivo studies have shown that this microorganism significantly reduces oxidative stress [5]. Phenolic compounds, including simple phenols, flavonoids, tannins, lignins, and phenolic acids and their derivatives, are potential candidates for scavenging free radicals due to their redox properties [6]. Phenolic acid molecules consist of benzene ring carboxylic groups and one or more hydroxyl or methoxy group, and the donation of electron or hydrogen atoms stabilizes the free radicals, conferring their antioxidant activity [1].

Commercial-scale algal cultivation has been occurring for over a decade [7], mainly to produce *Spirulina spp.* for natural high protein supplements, *Haematococcus spp.* as a source of the antioxidant astaxanthin, and *Dunaliella salina* for provitamin A production. Astaxanthin has high commercial value, and the production of natural astaxanthin from *Haematococcus spp.* has significantly greater antioxidant capacity than that of synthetic astaxanthin. Synthetic astaxanthin is sold at a lower price than that of natural astaxanthin; thus, the improved production of natural astaxanthin has attracted strong research interest. [8]. Recent studies have evaluated various strategies to increase important biomolecule synthesis by microalgae, such as lipids for biodiesel [9] and carbohydrates for bioethanol [10] production. The high cost of microalgae cultivation has been a major obstacle for commercializing its products. One of the viable solutions to reduce the costs of microalgae cultivation is to couple the high-value biomass production system for commercialization with agroindustry waste treatment since microalgae are known to effectively eliminate a variety of pollutants in wastewater, such as nitrogen, phosphates and organic carbons, among others [11]. Mixotrophic cultivation is a preferable microalgae cultivation technique for biomass production [12]. However, mixotrophic cultures are susceptible to contamination, which substantially influences microalgae growth, suggesting that closed and sterilized photobioreactors instead of open ponds should be used for cultivation. Despite the higher cost of photobioreactors, biomass yields can reach 5–15 g/L, which is 3–30 times higher than those produced under autotrophic growth conditions [13].

A polluting byproduct of the dairy industry in Brazil, which has environmental repercussions, is cheese whey, which is the residue from manufacturing various types of cheese, yogurt, ice cream and butter through different processes. Cheese is one of the world's leading agricultural products, with whey wastewater generation that is approximately four times the volume of processed milk [14]. It has been demonstrated that it is possible to produce microalgae biomass in mixotrophic cultures with whey since this waste provides important substrates, such as sugars, as a source of organic carbon for microalgae species [15].

Girard et al. [15] cultured *Scenedesmus obliquus* under mixotrophic conditions and demonstrated that this species presented higher specific growth rates and biomass yields when replacing 40% (v/v) of the standard medium with whey than when it was cultured with only standard medium. Salla et al. [2] studied the mixotrophic cultivation of *S. platensis* in diluted Zarrouk culture medium with the addition of concentrated whey protein residues and high lactose levels. They observed that there was an increase in biomass and carbohydrate

productivity by the species under study. Tsolcha et al. [16] developed a biological (algal) secondary cheese whey wastewater treatment system to produce biodiesel while simultaneously removing the polluting nutrients and chemical oxygen demand. Recent studies have demonstrated that *S. platensis* is also a promising bioethanol producer due to the carbohydrate concentration in its biomass, which could potentially reach 50% [2] and up to 60% [10] by changing the growth medium. In addition, the species may contain a high antioxidant compound content, which is capable of adding value to the cultivation process.

The abovementioned potential of *S. platensis* has motivated developing research to produce high-value biomass of commercial interest under mixotrophic growth conditions to reduce costs in a sustainable production system. Since whey is a potential pollutant produced in large volumes by dairy industries around the world, the aim of this research was to cultivate *S. platensis* under mixotrophic culture conditions using Zarrouk medium supplemented with whey from the production of buffalo mozzarella cheese to study its influence on the biomass production, specific growth rates, biochemical composition and antioxidant activity of this species.

## Materials and methods

### Microorganisms and culture medium

The *S. platensis* D9Z strain was obtained from the Microalgae Bank of the Laboratório de Ambientes Recifais e Biotecnologia com Microalgas (LARBIM) of the Federal University of Paraíba, Brazil. A xenic strain of *Spirulina platensis* was maintained under sterile conditions in test tubes at a temperature of 25°C and irradiance of 238 μmol m$^{-2}$ s$^{-1}$ under a 12 h light/dark photoperiod in Zarrouk standard culture medium [17]. For the mixotrophic cultures, the complete Zarrouk medium was supplemented with 2.5% (v/v), 5.0% (v/v) and 10% (v/v) buffalo mozzarella cheese whey from the TAPUIO Agropecuária Ltd. cheese industry located in Rio Grande do Norte, Brazil. The cultivation conditions used in this work, such as temperature, irradiance and aeration, were based on previous studies that describe the optimum conditions for *S. platensis* cultivation [18].

**Clarification of cheese whey for microalgae cultivation.** The whey was collected and stored in plastic bottles in a freezer at -20°C until its preparation for inoculation. The whey was clarified for use in the experiments. For use in the growth medium, the previously clarified whey was thawed in a refrigerator at 5°C and then autoclaved (121°C, 15 min), filtered through a 20 μm screen and centrifuged in a 24-BT simplex II centrifuge for 15 min at 1500 rpm for removal of the precipitated material. The whey was then autoclaved again to avoid possible contamination prior to addition into the growth medium. (dx.doi.org/10.17504/protocols.io.4t5gwq6)

### Experimental design

The inocula was obtained from cultures using Zarrouk medium [17]. Precultures (1 L) were made in Erlenmeyer flasks under the same conditions as that of the experimental apparatus to obtain the amount of cells needed to start the experiments. (dx.doi.org/10.17504/protocols.io.4txgwpn)

Cultures were prepared aseptically in a device with 5 fluorescent lamps of 45 W with a luminous intensity of 238 μmol m$^{-2}$ s$^{-1}$ measured on the outer surface of the vials using a Q201 quantum radiometer (Macam Photo-Metrics Ltd., Livingston, Scotland); a 12 h light/dark photoperiod was implemented under constant sterile aeration promoted by pumps (JAD Air Pump S-510) at a specific flow rate of 0.5 vvm (volume of air per volume of medium per minute). The initial *S. platensis* concentration ranged from 0.2 to 0.3 g/L. All assays were performed in triplicate. When the cultures reached the stationary phase, the biomasses were

collected by filtration on 20 μm screens and washed in distilled water to remove salt remnants. Then, the biomass was lyophilized in a lyophilizer (LJJ02—JJ Scientific) and frozen in a freezer at -20˚C until chemical characterization. The pH of the cultures was monitored every 24 hours throughout the duration of the experiment with a previously calibrated pH meter (K39-0014PA). (dx.doi.org/10.17504/protocols.io.4tygwpw)

## Determination of cellular concentration

The cell concentration was monitored daily until the cultures reached the stationary phase, a period of approximately 17 days until reaching the early stationary phase of growth, which is when carbohydrates are accumulated in the microalgal biomass [2]. The growth monitoring of the *S. platensis* cultures was performed by measuring the absorbance at λ = 670 nm in a spectrophotometer (SP-22 Biospectro). A calibration curve was generated to relate the absorbance to the cell dry weight. To quantify the biomass, a known volume of culture was filtered on 40 μm pore glass fiber filters (47 mm, Sartorius), and the obtained biomass was oven dried at 80˚C and quantified by weight difference according to Lourenço [19]. The filters used were previously treated in a muffle furnace at 400˚C for 4 h.

## Growth parameters

The growth rate ($\mu$ d$^{-1}$) of *S. platensis* was calculated as the slope of the natural log of the biomass concentration versus time during the exponential phase, when the correlation coefficients of these two variables were above 0.98 [20]. The duration of the exponential growth phase was different for each evaluated cultivation condition and was considered in the calculations.

The maximum biomass productivity $P_{Xmax}$ (g.L.day$^{-1}$) was evaluated according to Eq (1), where $X_{max}$ is the maximum biomass concentration at time t, Xo is the biomass concentration at time zero, and t is time.

$$P_{Xmax=} \frac{X_{max} - Xo}{t} \qquad (1)$$

The productivity was calculated for each day compared to that of day zero. The maximum value obtained was defined as the maximum biomass productivity.

## Chemical analyses

Clarified buffalo cheese whey samples were analyzed for the percentage of total solids, total protein, fat, lactose, and moisture using a DairySpec (Bentley Instruments Inc., Chaska Minnesota, USA). The equipment was calibrated using buffalo cheese whey calibration samples with different concentrations.

The dry matter and the ash content were quantified according to the AOAC [21] methodology. The total protein content was determined by the Kjeldahl method [21]. Kjeldahl nitrogen was measured (TECNAL digester block, model TE-040; TECNAL Kjeldahl TE-0364 distillation unit) using 200 mg samples of the dry biomass. The sample was first digested using a concentrated sulfuric acid digester solution following a progressive heating ramp and then distilled using boric acid, sodium hydroxide and indicator solutions. After distillation, the obtained solution was titrated with 0.1 N hydrochloric acid (HCl). The total protein content was calculated by multiplying the value of total nitrogen by a conversion factor of 4.78 [22]. Total carbohydrate determination was performed by a method published by Korchet [23] and adapted by Derner [24]. The standard curve was prepared from an anhydrous glucose solution. The total lipid content was determined according to Bligh and Dyer [25]. Extraction and

esterification of the fatty acids present in the biomass were carried out according to Menezes et al. [26] to identify the fatty acid profile.

A gas chromatograph (Thermo Scientific-CG/FID-FOCUS) with a flame ionization detector (FID) and a Supelco SP2560 capillary column (100 m × 0.25 mm, 0.2 μm) was used to separate the fatty acids. Nitrogen was used as the carrier gas (1.2 mL/min). The injector and detector temperatures were 230˚C and 270˚C, respectively. The column temperature programming was as follows: the temperature was held at 40˚C for 3 min, then increased to 180˚C for 5 min at a rate of 10˚C/min; the temperature was increased again to 220˚C for 3 min at a rate of 10˚C/min, and, finally, the temperature was increased to 240˚C at a rate of 20˚C/min and was maintained for 25 min. Nitrogen was used as the carrier gas at 0.9 mL/min. The injected sample volume was 1 μL with a split ratio of 10:1. The peaks were integrated and compared to a Supelco™37 Fatty Acid Methyl Ester (FAME) standard mixture from Sigma Aldrich.

## Antioxidant activity

The lyophilized microalgae biomass was extracted with methanol to obtain the antioxidant extracts. One gram of each sample was weighed, 40.0 mL of methanol was added, and the mixture was sonicated in an ice bath (Unique, model USC-1400A) for 20 min. For extraction, the sample was shaken for three hours at 25˚C and centrifuged at 8000 rpm for 10 min. The extracts were dried in an air circulating oven (SOLAB, model SL102) at 35˚C and stored in the dark at room temperature until they were used. Each extract was resuspended to 5.0 mg/mL with distilled water and stored in amber bottles under refrigeration until use. The entire extraction procedure was performed under in 4.8 lux illumination to avoid photodegradation.

## Phenolic compounds

The total extractable phenolic contents were determined by the Folin-Ciocalteu colorimetric method [27]. The concentration of phenolic compounds was estimated using a standard calibration curve with gallic acid. The results were expressed as mg GAE/g (milligrams of gallic acid equivalent per gram of extract).

## Antioxidant capacity via the ABTS method

The antioxidant capacity of the microalgae via the ABTS•+(*2,2′-azino-bis(3-ethyl benzothiazoline-6-sulfonic acid*) method was determined according to the methodology described by Rufino et al. [28]. The standard curve was prepared using Trolox at μmol concentrations. The antioxidant capacity results were expressed in μmol Trolox/g extract (antioxidant capacity equivalent to that of Trolox).

## Antioxidant capacity via the FRAP assay

The antioxidant capacity of the microalgae extracts was estimated by the iron reduction method (FRAP), which is based on the ability of an antioxidant to reduce $Fe^{3+}$ to $Fe^{2+}$ and was performed following the methodology described by Benzie and Strain [29]. Two standard curves were prepared, one using Trolox and one with ferrous sulfate. The antioxidant capacity results were expressed in μmol Trolox/g extract (antioxidant capacity equivalent to that of Trolox) or μmol ferrous sulfate/g extract (antioxidant capacity equivalent to that of ferrous sulfate).

## Statistical analysis

The data were utilized to calculate the average value of three independent tests, and the values were expressed as the mean ± standard deviation (SD). Statistical analyses were performed with SAS software version 9.0. The data were analyzed using a general linear model (PROC MEANS and GLM). Duncan's new multiple range test was used to compare treatment averages (whey inclusion levels) of the microalgae characteristics. Statistical significance occurred in all analyses when the calculated p-value was ≤0.05.

## Results and discussion

### Whey composition

The whey composition of buffalo mozzarella cheese may vary depending on several factors, including the quality, milk composition and technology of the manufacturing process [30]. According to Sales et al. [30], whey represents approximately 55% of the nutrients present in the milk, thereby constituting a waste product that is rich in organic matter with great potential for mixotrophic and heterotrophic microalgae cultivation. The basic characterization of the clarified buffalo cheese whey is shown in Table 1.

Lactose represented almost 75% of the total solid percentage in the whey composition, constituting the main source of available carbon, along with the presence of proteins and fats in smaller proportions.

### Growth curve of *S. platensis*

*S. platensis* was grown under autotrophic and mixotrophic conditions with 2.5%, 5.0% and 10.0% cheese whey for 17 consecutive days until reaching the early stationary phase of growth to evaluate important kinetic parameters, such as maximum biomass concentration, maximum yield and specific growth rate. The whey concentrations utilized were based on previous studies. Mouther [31] showed that the optimal cheese whey concentration for *S. platensis* mixotrophic cultivation was 3.0%, and inhibition occurs at higher whey concentrations. Moreover, heterotrophic cultivation was inappropriate for *S. platensis*, possibly due to some inhibiting factors for this species present in the whey. Fig 1 shows the growth curves of *S. platensis* in the cultures evaluated in this study:

The lag phase for all cultures occurred within the first 24 hours. This is expected since inoculum was used in the standard growth medium. The maximum biomass production for all cultures was obtained at the early stationary phase of growth. The control culture presented a maximum biomass concentration of 1.75 g/L, which was the lowest value when compared to that of all the studied conditions (Table 2). It was observed that autotrophic $X_{max}$ was similar to the values found by Kumari et al. [32] (1.87 g/L) and superior to the values found by Andrade and Costa [33] (1.44 g/L) and Mourthé [31] (1.01 g/L). Mixotrophic cultivation of *S. platensis* with 5.0% whey reached the highest biomass yield (2.98 g/L). Similar values were

**Table 1. Composition of clarified buffalo mozzarella cheese whey used for the cultivation of *S. platensis*.**

| Variables | | Clarified cheese whey | |
|---|---|---|---|
| Total solids (%) | | 6.77 | |
| | Fat (%) | | 0.02 |
| | Total Protein (%) | | 0.60 |
| | Lactose (%) | | 5.07 |
| Moisture (%) | | 93.23 | |

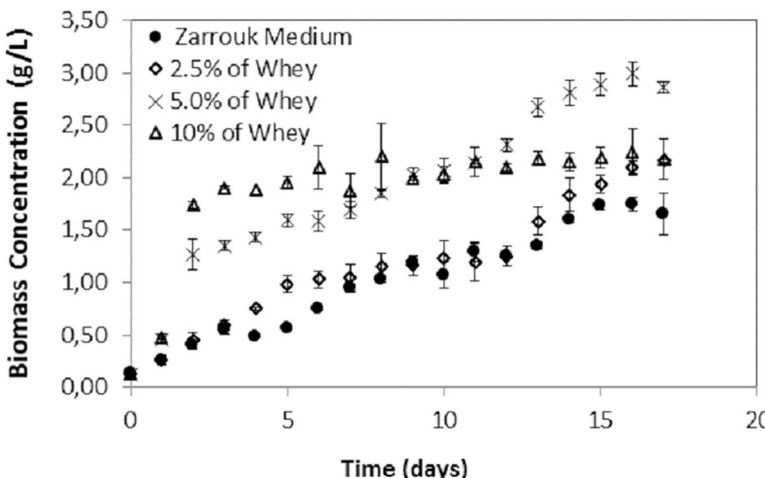

**Fig 1. Growth curve of *S. platensis* grown on autotrophic Zarrouk medium and mixotrophic medium supplemented with buffalo mozzarella cheese whey at concentrations of 2.5%, 5.0% and 10.0%.** The error bars represent the standard deviations (n = 3).

observed by Salla et al. [2] in *S. platensis* cultures using whey protein concentrate. This value was also similar to that found by Andrade and Costa [33] (2.8 g/L) when cultivating *S. platensis* in diluted Zarrouk medium with added molasses, a byproduct from the sugarcane industry, and subtly higher than the values reported by Marquez et al. [34] (2.5 g/L) and Chen and Zhang [35] (2.4 g/L) when they used glucose as an organic substrate. The culture with 10.0% whey showed double the specific growth rate of the culture with 5.0%, and it had the highest biomass productivity on the 4th day (Fig 2).

Mixotrophic cultivation of *S. platensis* using 2.5% and 5.0% whey was favorable for biomass yield. This process could reduce the cost of producing microalgae while also consuming a waste that is discarded into the environment. The growth medium supplemented with 10.0% whey would be relevant in an industrial process since the biomass could be produced in the largest amount in a short time interval of approximately 4 days of growth. It is very important to evaluate the biochemical composition of the biomass in this period to check the viability of producing certain metabolites of commercial interest, which will be done at a later stage. Currently, many studies involving microalgae culture in effluents have been researched mainly as a strategy for lipid and carbohydrate production [2] [9]. The biomass yields and growth rates of *S. platensis* observed in this study were higher than those observed in the studies reported by Tsolcha et al. [16], Economou et al. [36] and Dourou et al. [37]. All of these microalgae cultivations were performed under non-aseptic conditions that, despite minimizing the cost of the process, led to lower biomass yields. Mixotrophic and heterotrophic cultures under aseptic

**Table 2. Maximum biomass concentration ($X_{max}$), maximum yield ($Y_{max}$) and specific growth rate (μ) of *S. platensis*.**

| Cheese Whey % | $X_{max}$ (g.L$^{-1}$) | $P_{max}$ (g.L$^{-1}$.day$^{-1}$) | μ (day$^{-1}$) |
|---|---|---|---|
| 0.0 | 1.65[c]±0.18 | 0.1011[b]±0.003 | 0.183[d]±0.008 |
| 2.5 | 2.16[b]±0.02 | 0.1195[b]±0.001 | 0.258[c]±0.013 |
| 5.0 | 2.98[a]±0.07 | 0.1793[a]±0.005 | 1.024[b]±0.018 |
| 10.0 | 2.18[b]±0.16 | 0.1628[a]±0.015 | 2.048[a]±0.036 |

Different letters indicate significant differences for different treatments (p < 0.05).

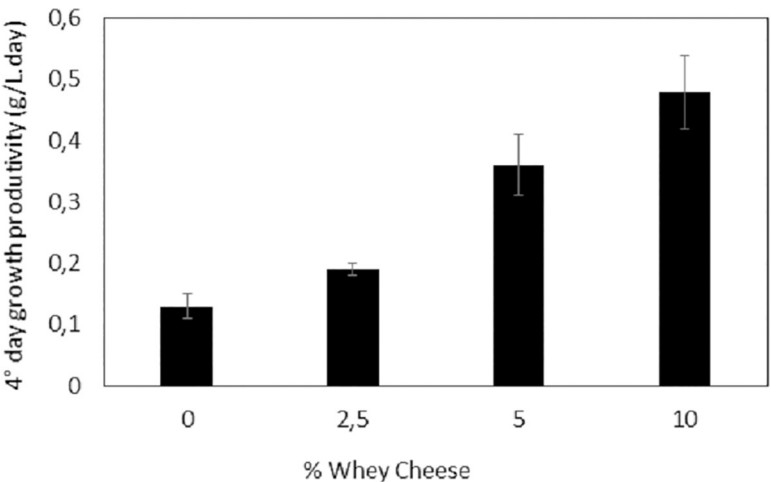

**Fig 2. Productivity of *S. platensis* grown on autotrophic Zarrouk medium and mixotrophic medium supplemented with buffalo mozzarella cheese whey on the 4th day of growth.** The error bars represent the standard deviations (n = 3).

conditions can be developed in closed photobioreactors, leading to high biomass and/or bio-compound yields, which may justify the higher cost of making the process feasible. Mixo-trophic cultures show reduced photoinhibition and higher growth rates in relation to those of autotrophic and heterotrophic cultures, having independence in assimilating both growth sub-strates and the ability to perform photosynthesis as an advantage since the acetyl-CoA pool is maintained for both carbon sources of $CO_2$ fixation (Calvin cycle) and for the extracellular organic carbon [38].

## pH control

The pH influences the $CO_2$ and mineral solubility in the growth medium, directly or indirectly interfering with the algae metabolism; thus, it is one of the most important growth parameters. The pH of the cultures during growth was monitored every 24 h, as shown in Fig 3:

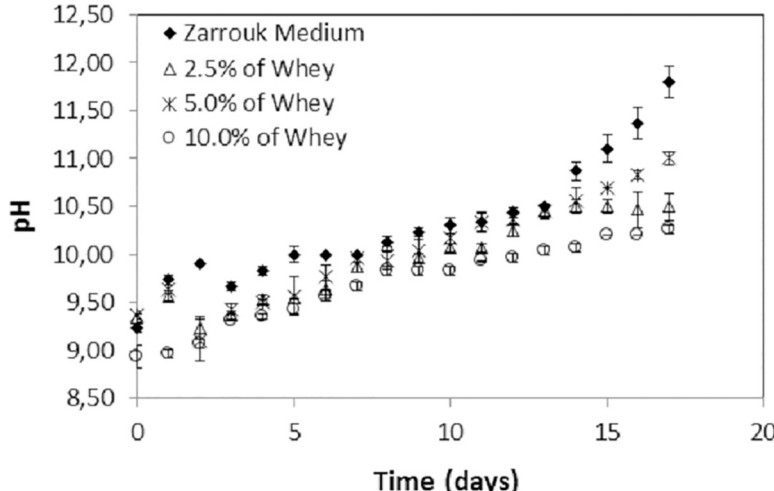

**Fig 3. pH, which was monitored every 24 hours in *S. platensis* cultures grown in Zarrouk autotrophic medium and mixotrophic medium supplemented with buffalo mozzarella cheese whey.** The error bars represent the standard deviations (n = 3).

Mixotrophic cultures had lower pH ranges than that of the control culture. This is expected since the whey is acidic. The pH values in these cultures showed similar ranges from the beginning to the end of the experiment, where the lowest value was observed in the culture with 10.0% whey. In general, pH variation was more pronounced for the control (9.3–11.8) with a notable increase in the last 3 days. The autotrophic growth medium is usually rich in bicarbonate, and inorganic carbon may be in the form of $CO_2$, $H_2CO_3$, $HCO_3^-$ (bicarbonate) or $CO_3^-$ (carbonate), and their proportions depend on pH. The proportions of $HCO_3^-$ or $CO_2^-$ increase at basic pH, but $CO_2$ is the carbon source that is predominantly used by algae when the pH of the growth medium is low. The growth of the microalgal cultures was confirmed by the gradual increase in the pH of the medium, which is attributed to the inorganic carbon consumption for cell growth. Inorganic carbon consumption forces a displacement of the carbonate-bicarbonate equilibrium in the buffer system [39]. The optimal pH growth range of *S. platensis* is 9.5 to 10.5 [40]. Although the pH was not in the optimal working range, there was no growth limitation when comparing the biomass production and growth rate of *S. platensis* cultivated in this work with those of other studies reported in the literature, as discussed above.

## Biochemical composition

*S. platensis* is one of the most studied microalgae species and is widely used as a food product and dietary supplement due to its nutritional profile and high nutrient bioavailability. According to Habib et al. [41], the basic chemical composition of *S. platensis* consists of proteins (50–70%), carbohydrates (15–25%), lipids (6–8%) and minerals (7–13%), and these percentages may vary depending on the type of culture. This species has also been studied in the production of biofuels, mainly bioethanol and bio-oil, because it is a low lipid species and is an attractive biomass for thermal conversion and fermentation processes [16] [42] [43]. The biochemical composition of the biomass obtained in this work is presented in Table 3.

The biomass obtained from autotrophic cultivation presented a high total protein content and low carbohydrate, lipid and ash contents, similar to the results reported by Madkour et al. [44] and Salla et al. [2]. The ash content increased in the cultures with the addition of 2.5% and 5.0% whey. Madkour et al. [44] demonstrated that the protein and carbohydrate contents for *S. platensis* grown in Zarrouk growth medium were 52.95% and 13.20%, respectively. Salla et al. [2] obtained a carbohydrate content ranging from 20.0% to 60.0% and a protein content of 45.40%. The biochemical composition of microalgae is strongly influenced by the growth medium, temperature, aeration, irradiance, and culture volume, among other factors [45]. In this study, all variables were held constant, and only the growth medium composition changed, so this parameter was responsible for the observed differences in the biochemical composition. It is possible to notice a clear, direct relationship in which the whey content increased in the growth medium and the total protein synthesis decreased for all studied

**Table 3. Biochemical composition of *S. platensis* biomass cultivated in autotrophic growth medium and mixotrophic growth medium with the addition of buffalo mozzarella cheese whey.**

| Cheese Whey (%) | Dry Matter (%) | Total Protein (%) | Fat (%) | Carbohydrates (%) | Ash (%) |
|---|---|---|---|---|---|
| *0.0* | 93.65[ab]±1.60 | 65.55[a]±1.88 | 5.18[a]±0.38 | 27.17[c]±1.57 | 6.04[b]±0.38 |
| *2.5* | 91.77[bc]±0.6 | 60.62[b]±0.30 | 2.04[d]±0.03 | 23.29[c]±1.54 | 14.21[a]±0.03 |
| *5.0* | 90.57[c]±0.52 | 44.56[c]±0.74 | 2.56[c]±0.15 | 47.83[a]±0.78 | 14.03[a]±0.14 |
| *10.0* | 94.92[a]±0.99 | 39.62[d]±0.36 | 3.40[b]±0.18 | 40.65[b]±3.66 | 6.36[b]±012 |

Different letters indicate significant differences for different treatments ($p < 0.05$).

conditions, while the carbohydrate content increased in the cultures with whey additions above 2.5%, according to Fig 4.

There was an increase in carbohydrates and a decrease in protein content in the mixotrophic cultures. *S. platensis* produces high levels of protein, which justifies its wide commercial application as a protein-rich supplement, and this mainly occurs when the growth medium is rich in nitrogen. It has been shown that some types of microalgae accumulate carbohydrates and other non-nitrogenous biocompounds when an organic source is added to the growth medium, so these results are expected [2]. This study demonstrated the feasibility of applying mixotrophic cultivation to produce high-carbohydrate microalgae. Moreover, *S. platensis* is a species that has been studied in thermal degradation processes for producing bio-oil, biochar and gas. Some studies have shown that bio-oils produced from microalgae are more stable, have a lower oxygen content and a higher calorific value than that of lignocellulosic bio-oil [46]. However, bio-oil presents a high nitrogen content that is undesirable because it deactivates acid catalysts used for coprocessing crude oil in refineries and emits NOx during combustion [42]. In addition to increasing biomass production, the mixotrophic culture of *S. platensis* using cheese whey may decrease the processing cost and produce low protein biomass, thus being favorable for application in thermal degradation processes for producing biofuels.

*S. platensis* is a low lipid species when grown under standard conditions, but this percentage may increase as a function of culture conditions. It was observed that there was a decrease in

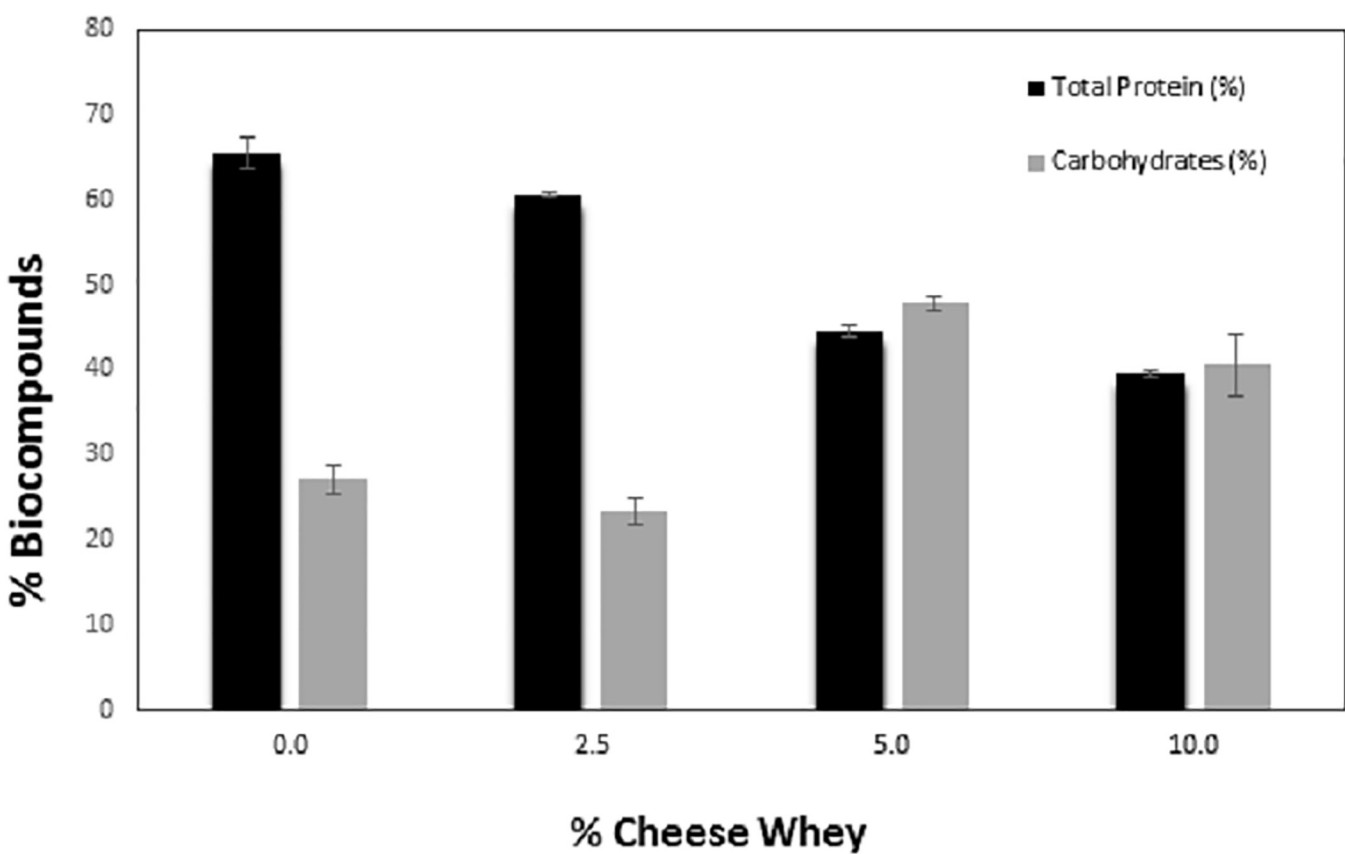

**Fig 4. Influence of the whey percentage on the synthesis of carbohydrates and proteins by *S. platensis*.** The error bars represent the standard deviations (n = 3).

**Table 4. The fatty acid methyl ester (FAME) profiles (percentage of total FAMEs) present in *S. platensis* biomass cultured in autotrophic medium and mixotrophic medium with the addition of buffalo mozzarella cheese whey.**

| Fatty Acid | Whey % | | | |
|---|---|---|---|---|
| | 0.0 | 2.5 | 5.0 | 10.0 |
| C10:0[1] | nd | nd | 0.70–0.73 | 0.62–0.69 |
| C12:0[2] | 0.97–1.11 | 0.73–0.75 | 0.60–0.64 | 0.62–0.65 |
| C13:0[3] | 1.43–1.51 | 1.01–1.03 | 0.74–0.92 | 0.92–0.93 |
| C14:0[4] | 0.62–1.39 | 0.71–0.72 | 0.59–0.59 | 0.23–0.24 |
| C14:1[5] | 0.89–0.96 | 0.69–0.69 | 0.48–0.48 | 0.50–0.51 |
| C15:0[6] | 0.23–0.55 | 0.32–0.33 | 0.28–0.30 | 0.30–0.31 |
| C15:1[7] | 0.66–0.68 | 0.62–0.63 | 0.58–0.60 | 0.50–0.67 |
| C16:0[8] | 40.60–42.15 | 45.72–45.75 | 44.87–44.91 | 44.29–43.68 |
| C16:1[9] | 1.67–1.76 | 2.04–2.23 | 2.29–2.31 | 2.82–2.87 |
| C17:0[10] | 0.16–0.27 | 0.49–0.49 | 0.18–0.35 | 0.16–0.35 |
| C17:1[11] | 0.25–0.28 | 0.22–0.23 | nd | 2.41–2.51 |
| C18:0[12] | 2.31–4.42 | 2.28–2.39 | 2.45–2.55 | 0.24–0.25 |
| C18:1n9c[13] | nd | 0.63–0.67 | 069–0.74 | 0.68–5.53 |
| C18:1n9t[14] | 9.78–11.38 | 10.92–10.92 | 12.00–12.06 | 3.25–3.26 |
| C18:2n6c[15] | nd | 0,34–0,36 | nd | nd |
| C18:2n6t[16] | 10.99–11.59 | 10.46–10,45 | 8.84–8.87 | 10.68–11.24 |
| C20:0[17] | 14.54–15.15 | 16.09–16.13 | 17.25–17.29 | 19.88–19.89 |
| C20:2[18] | 0.16–0.19 | nd | nd | nd |
| C20:3n6[19] | 0.57–0.60 | 0.26–0.21 | 0.43–0.44 | nd |

[1]Capric acid

[2]Lauric acid

[3]Tridecanoic acid

[4]Myristic acid

[5]Myristoleic acid

[6]Pentadecanoic acid

[7]Cis-10-pentadecanoic acid

[8]palmitic acid

[9]Palmitoleic acid

[10]Heptadecanoic acid

[11]Cis-10-heptadecanoic acid

[12]Stearic acid

[13]Oleic acid

[14]Elaidic acid

[15]Linoleic acid

[16]Linolelaidic acid

[17]Arachidic acid

[18]Cis-11,14-eicosadienoic acid

[19]Cis-8,11,14-eicosatrienoic acid

nd: not detected

(These numbers are the lowest and highest FAME contents).

lipid synthesis in the mixotrophic cultures of *S. platensis* when compared to that of the control culture, but there were no significant differences as a function of the added whey percentage. The fatty acid methyl ester (FAME) profiles of *S. platensis* are shown in Table 4.

There were no significant changes in the FAME profile, except for the culture with 10.0% whey, in which the percentage of C18:1n9c decreased by 70%. The fatty acid ester found at the greatest percentage was C16:0 (40.60–45.75%), followed by C20:0, C18:2n6c and C18:1n9c. Prates et al. [47] reported that palmitic acid was predominant in the fatty acid profile of *S. platensis*. This acid is an important source of energy in infant feeding since breast milk contains 20% to 30% of this fatty acid. However, saturated fatty acids have been associated with an increased risk of cardiovascular disease in adults [48]. Palmitic acid is widely used in the pharmaceutical, cosmetic and surfactant industries and can be successfully applied industrially [49].

## Antioxidant activity

The total phenolic content and antioxidant activity of the *S. platensis* biomass were quantified as shown in Tables 5 and 6:

*S. platensis* presented a high phenolic compound content (22.37–33.23 mg gallic acid/g) when compared to that reported in other studies by Sousa and Silva [50] (1.65 mg/g), Parisi et al. [51] (4.22–4.41 mg/g), and Colla et al. [52] (0.0024–0.0049 mg/g). There were no significant differences in the phenolic compound concentrations when autotrophic and mixotrophic cultures were compared, except for a reduction of 33.0% in the culture with 10.0% whey. The phenolic compound production in this study was probably potentiated as a function of the irradiance of the cultures (238 $\mu mol\ m^{-2}\ s^{-1}$), according to a study conducted by Kepekçi and Saygideger [53]. Although there are no previous studies reporting the influence of mixotrophic cultivation on microalgae antioxidant synthesis, some studies have reported that environmental stresses such as exposure to metals [45] or stress by exposure to UV light [54] increased the synthesis of phenolic compounds.

The production of antioxidants depends on the cultivation conditions and the extraction method for quantification. In general, the antioxidant activity of the mixotrophic cultures decreased when compared to that of the autotrophic culture. The antioxidative activity in the autotrophic culture increased in the following order: $FRAP_{sf}>FRAP_{Tx}>ABTS^{\bullet+}$, while the same performance was observed for all assays in the mixotrophic cultures, regardless of the whey content in the growth medium; therefore, the antioxidant activity was similar when evaluated by $ABTS^{\bullet+}$ and $FRAP_{sf}$ and less when evaluated by the $FRAP_{tx}$ method. The antioxidant activity of *S. platensis* obtained by the $ABTS^{\bullet+}$ method (11.68–17.67 µmol Tx/g) was lower than the values reported by Kepekçi & Saygideger [53] (31.41–54.16 µmol Trolox $g^{-1}$). The value observed with the $FRAP_{tx}$ method from the autotrophic culture (20.68 µmol Tx/g) was higher than that found by Hossain et al. [55] (8.81 µmol Tx/g). It is likely that the inhibition of oxidative stress resulted in the generation of free radical species in the cells, which reacted and decreased antioxidant synthesis, possibly because the medium provides favorable conditions for the organisms and has no influence on promoting antioxidant production [3]. Goiris et al. [3] observed a reduction in the antioxidants of microalgae biomass when nitrogen-limited medium was used. They concluded that nutritional stress is not an effective strategy for improving the overall antioxidant content in microalgae.

**Table 5. Total extractable phenolic content.**

| Biomass | % Whey | | | |
|---|---|---|---|---|
| | 0.0 | 2.5 | 5.0 | 10.0 |
| Phenolic compounds (mg gallic acid/g) | 33.20[a]±0.92 | 30.60[b]±0.43 | 33.23[a]±0.36 | 22.37[c]±0.14 |

Different letters indicate significant differences for different treatments (p < 0.05).

**Table 6. Antioxidant activity of *S. platensis* by the ABST•+ (2,2'-azino-bis (3-ethylbenzothiazolin)-6-sulfonic acid) free radical scavenging method and by FRAP ferric reduction.**

| Whey % | FRAP$_{sf}$ (µmol FS/g) | FRAP$_{tx}$ (µmol Tx/g) | ABTS•+ (µmol Tx/g) |
|---|---|---|---|
| 0.0 | 35.35[a]±0.87 | 20.68[a]±0.51 | 17.67[a]±0.52 |
| 2.5 | 15.25[b]±0.08 | 8.95[b]±0.05 | 14.73[b]±0.14 |
| 5.0 | 15.39[b]±0.10 | 9.03[b]±0,06 | 14.62[b]±0.45 |
| 10.0 | 14.54[b]±0.14 | 8.54[b]±0,08 | 11.64[c]±0.14 |

FRAP (*ferric reducing antioxidant power;* ABTS•+ (2,2'-azino-bis(3-ethylbenzothiazolin)-6-sulfonic acid). Different letters indicate significant differences for different treatments ($p < 0.05$).

The results of the present study support that the mixotrophic cultivation of *S. platensis* with 5% whey could be a viable strategy for the production of phenolic compounds and/or antioxidant compounds with a relatively low cost-optimized process since the biomass productivity was higher in this condition than at lower whey cultivation conditions. No antioxidant activity data were found in the literature in similar culture conditions as performed in this study. Many studies have evaluated the antioxidant capacity of foods supplemented with *S. platensis*, such as wheat bread dough [48], cookies [49] and yogurt [56], among others. They observed that the antioxidant capacity of all foods was higher when supplemented with *S. platensis*.

## Conclusions

The biomass production, biochemical composition and antioxidant capacity of the mixotrophic *S. platensis* microalgae cultured in growth medium with different cheese whey contents were evaluated. In general, the mixotrophic cultivation increased the biomass and carbohydrate productivity and decreased the antioxidant capacity, protein and lipid productivity; however, there were no significant differences in phenolic compound content between the cultivation conditions, except for that of the culture with 10.0% whey. The maximum cell concentration (2.98 g/L) was obtained in the experiment with the addition of 5.0% whey. There was a 70% and 76% increase in biomass production and carbohydrate content, respectively when compared to those of the control. Under these culture conditions, the increase in carbohydrate production in *S. platensis* indicated that this species is a potential biomass source for the production of bioethanol. These data help to explain the microalgal growth and its chemical composition and enhance our understanding of how to improve the performance of mixotrophic cultivation. Overall, the results support the mixotrophic cultivation of *S. platensis* as a viable strategy to reduce the production costs of biomass and carbohydrates and simultaneously contribute to mitigating the environmental problems caused by eliminating whey in the dairy industry. If the cost of microalgae production can be optimized, bioethanol might be obtained in large-scale production combined with wastewater treatment and carbon sequestration.

## Supporting information

**S1 File. Supporting information.**
(DOCX)

## Acknowledgments

The authors gratefully acknowledge the Laboratório de Nutrição Animal from the Universidade Federal do Rio Grande do Norte for chemical analysis and the Laboratório de Ambientes

Recifais e Biotecnologia com Microalgas from the Universidade Federal da Paraíba for biochemical analysis and for the use of their facilities.

## Author Contributions

**Conceptualization:** Emanuelle P. E. Silva.

**Data curation:** Maria I. B. Pereira, Bruna M. E. Chagas, Adriano H. N. Rangel.

**Formal analysis:** Maria I. B. Pereira, Bruna M. E. Chagas, Guilherme F. Medeiros, Emerson M. Aguiar, Adriano H. N. Rangel.

**Funding acquisition:** Bruna M. E. Chagas, Roberto Sassi, Guilherme F. Medeiros, Emerson M. Aguiar, Júlio C. Andrade Neto, Adriano H. N. Rangel.

**Investigation:** Maria I. B. Pereira, Bruna M. E. Chagas, Emanuelle P. E. Silva, Adriano H. N. Rangel.

**Methodology:** Maria I. B. Pereira, Bruna M. E. Chagas, Roberto Sassi, Guilherme F. Medeiros, Emerson M. Aguiar, Emanuelle P. E. Silva, Adriano H. N. Rangel.

**Project administration:** Bruna M. E. Chagas, Adriano H. N. Rangel.

**Resources:** Júlio C. Andrade Neto, Adriano H. N. Rangel.

**Software:** Luiz H. F. Borba.

**Supervision:** Bruna M. E. Chagas, Roberto Sassi, Guilherme F. Medeiros, Emerson M. Aguiar, Júlio C. Andrade Neto.

**Validation:** Bruna M. E. Chagas, Luiz H. F. Borba, Adriano H. N. Rangel.

**Writing – original draft:** Bruna M. E. Chagas, Emanuelle P. E. Silva, Adriano H. N. Rangel.

**Writing – review & editing:** Maria I. B. Pereira, Bruna M. E. Chagas, Roberto Sassi, Guilherme F. Medeiros, Emerson M. Aguiar, Luiz H. F. Borba, Júlio C. Andrade Neto, Adriano H. N. Rangel.

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
