## [Editor Report · Decision Letter 0]

12 Jun 2019

PONE-D-19-15520

Mixotrophic cultivation of Spirulina platensis in dairy wastewater: effect on the production of biomass, biochemical composition and antioxidant capacity

PLOS ONE

Dear Dr. Emerenciano,

Thank you for submitting your manuscript to PLOS ONE. After careful consideration, we feel that it has merit but does not fully meet PLOS ONE’s publication criteria as it currently stands. Therefore, we invite you to submit a revised version of the manuscript that addresses the points raised during the review process.

Please see the comments stated as Additional Editor Comments.

We would appreciate receiving your revised manuscript by Jul 27 2019 11:59PM. To enhance the reproducibility of your results, we recommend that if applicable you deposit your laboratory protocols in protocols.io, where a protocol can be assigned its own identifier (DOI) such that it can be cited independently in the future. For instructions see: http://journals.plos.org/plosone/s/submission-guidelines#loc-laboratory-protocols

We look forward to receiving your revised manuscript.

Kind regards,

Branislav T. Šiler

Academic Editor

PLOS ONE

Journal Requirements:

2. Thank you for stating the following in the Acknowledgments Section of your manuscript:"The authors gratefully acknowledge the CAPES (Coordenação de Aperfeiçoamento de Pessoal de Nível Superior) for financial support and Laboratório de Nutrição Animal from the Federal University of Rio Grande do Norte (EAJ/UFRN) for chemical analysis and LARBIM (Laboratório de Ambientes Recifais e Biotecnologia com Microalgas) from the

Federal University of Paraíba for biochemical analysis and for the facilities."

Please remove any funding-related text from the manuscript and let us know how you would like to update your Funding Statement. Currently, your Funding Statement reads as follows:"The funders had no role in study design, data collection and analysis, decision to publish, or preparation of the manuscript."

Additional Editor Comments:

The manuscript has no page neither line numbers. Reviewers have difficulties to track and manage the manuscript if page and line numbers do not exist. The text has too many paragraphs, especially in the Introduction section. Can you please compact it? I also noticed excessive number of references.

It is a common practice to use abbreviation of the species genus, such as S. platensis, once you already give the full species binomial name in the first mention.
---

## [Author Response · Author response to Decision Letter 0]

10 Jul 2019

3. Additional Editor Comments:

3.1. The manuscript has no page neither line numbers. Reviewers have difficulties to track and manage the manuscript if page and line numbers do not exist. 

 We agree with no page neither line numbers in the manuscript may be confusing for the reviewers. We added page and line numbers to the manuscript.

3.2. The text has too many paragraphs, especially in the Introduction section. Can you please compact it? I also noticed excessive number of references.

We agree with the text has too many paragraphs and references in the manuscript. We reduced paragraphs from the Introduction and Results and Discussion. The references were reduced from 75 to 59. 

3.3. It is a common practice to use abbreviation of the species genus, such as S. platensis, once you already give the full species binomial name in the first mention.

The name Spirulina platensis was abbreviated to S.platensis in the text body of the manuscript.

3.4. While revising your submission, please upload your figure files to the Preflight Analysis and Conversion Engine (PACE) digital diagnostic tool, https://pacev2.apexcovantage.com/. PACE helps ensure that figures meet PLOS requirements. To use PACE, you must first register as a user. Registration is free. Then, login and navigate to the UPLOAD tab, where you will find detailed instructions on how to use the tool. If you encounter any issues or have any questions when using PACE, please email us at figures@plos.org. Please note that Supporting Information files do not need this step.

We uploaded our figure files to the Preflight Analysis and Conversion Engine (PACE) digital diagnostic tool as suggested.

---

## [Decision Letter · Decision Letter 1]

30 Jul 2019

PONE-D-19-15520R1

Mixotrophic cultivation of Spirulina platensis in dairy wastewater: effect on the production of biomass, biochemical composition and antioxidant capacity

PLOS ONE

Dear Dr. Emerenciano,

Thank you for submitting your manuscript to PLOS ONE. After careful consideration, we feel that it has merit but does not fully meet PLOS ONE’s publication criteria as it currently stands. Therefore, we invite you to submit a revised version of the manuscript that addresses the points raised during the review process.

Reviewers raised major concerns about the general manuscript structure as well as the statistical significance of the presented differences in the results section. More comments are listed below.

We would appreciate receiving your revised manuscript by Sep 13 2019 11:59PM. To enhance the reproducibility of your results, we recommend that if applicable you deposit your laboratory protocols in protocols.io, where a protocol can be assigned its own identifier (DOI) such that it can be cited independently in the future. For instructions see: http://journals.plos.org/plosone/s/submission-guidelines#loc-laboratory-protocols

We look forward to receiving your revised manuscript.

Kind regards,

Branislav T. Šiler

Academic Editor

PLOS ONE

Reviewers' comments:

Reviewer's Responses to Questions

**Comments to the Author**

1. If the authors have adequately addressed your comments raised in a previous round of review and you feel that this manuscript is now acceptable for publication, you may indicate that here to bypass the “Comments to the Author” section, enter your conflict of interest statement in the “Confidential to Editor” section, and submit your "Accept" recommendation.

Reviewer #1: (No Response)

Reviewer #2: (No Response)

2. Is the manuscript technically sound, and do the data support the conclusions?

Reviewer #1: Partly

Reviewer #2: Yes

3. Has the statistical analysis been performed appropriately and rigorously? 

Reviewer #1: Yes

Reviewer #2: No

4. Have the authors made all data underlying the findings in their manuscript fully available?

Reviewer #1: Yes

Reviewer #2: No

5. Is the manuscript presented in an intelligible fashion and written in standard English?

Reviewer #1: No

Reviewer #2: No

6. Review Comments to the Author

Reviewer #1: Highlight: Not provided

Graphical abstract: Not provided

General remarks: This manuscript focus on the growth of S. platensis microalgae using a pollutant by-product called cheese whey for high biomass and carbohydrate production. The used of English in this manuscript is poor and require to further amend the language. There is a lot of work needed to further improve this manuscript (e.g., structure arrangement, language, unnecessary introduction) My recommendation is not to accept this paper for now as there is still lack of assurance and required amendment in this manuscript. I hope my following comments world improve the following manuscript.

COMMENTS:

The used of English and grammar were poor. This section is not constructive enough in this section

- Page 3, Line 63 – 65. Please read and cite some suggested paper for these biomolecules.

1. Khoo, K. S., Lee, S. Y., Ooi, C. W., Fu, X., Miao, X., Ling, T. C., & Show, P. L. (2019). Recent advances in biorefinery of astaxanthin from Haematococcus pluvialis. Bioresource technology, 121606.

2. Ren, H. Y., Xiao, R. N., Kong, F., Zhao, L., Xing, D., Ma, J., ... & Liu, B. F. (2019). Enhanced biomass and lipid accumulation of mixotrophic microalgae by using low-strength ultrasonic stimulation. Bioresource technology, 272, 606-610.

Also, please remove “bioactive compounds”.

- Page 4. Line 90 – 104, Please combine all into one paragraph.

-Why is such concentration (2.5%, 5.0% and 10%) of cheese whey deployed for the culture medium? Why not (2.5%, 5.0% 7.5% and 10%?). Please give clarification for this. By doubling up the concentration is difficult for optimization.

- Page 9, Line 266 – 269, the English used is too poor, please rephrase.

- Based on Figure 1, why did the author stop at Day 17 and state it’s the maximum growth? Why not increase to Day 20 to see if there is any growth?

- There is a lot of new paragraph for each section. Advise author to keep 1-3 paragraph is sufficient enough.

Page 10, Line 295 – 298, This paragraph does not make any sense and clueless on discussion. Please revise.

- I realize that the discussion part is too long and will get lost during reading. Advise author to revise overall of this manuscript. Keep it short and simple for the reasoning.

- Page 13, Line 376, what species? Please state it.

Page 17, Line 462 – 488. Combine all into one paragraph or remove it. It is too lengthy and not necessary for this result and discussion section. Much more to introduction.

- Please revise the structure of the overall manuscript. There are still too many new opening paragraphs which does not seem to be important.

- For Figure 2, why is the error bar for 10% whey cheese is so large. Some were small and some were large. This shows the inconsistency for the data collected.

Reviewer #2: The authors used an experimental approach and determined the biochemical composition of the algae, in addition to biomass yields, specific growth rates, and antioxidant activity. The authors compared their results to those found in other studies, providing plenty of context. They found that Spirulina platensis can reach higher biomass yields and specific growth rates when clarified cheese whey is added to the growth medium, and that carbohydrate content increases. Protein, lipid, and antioxidant contents decreased, however, which affects which products can be made from the algae. Overall the cheese whey would be effective for improving biomass production and reducing cultivation costs for algae products that need high carbohydrate costs.

One major way the study could be improved is by performing statistical analyses, which are not described in the methods section. However, results throughout the abstract, results, and discussion sections are often described as significant, even though it appears statistical tests were not performed. If statistical analyses are not going to be performed, the authors should be transparent about this, and only use qualitative descriptors to present the results. It appears that all data underlying results of the study are not available in the supplementary information (e.g., data for each of the control and treatment replicates that are used to provide the average and standard deviations in tables and figures). The authors should further clarify how their current manuscript complies with the journal’s data availability policy, or should include more of the experimental data in supporting documents or ideally in an online public data repository (e.g., Figshare). Lastly, the intended meaning of the manuscript writing is mostly understandable, but further editing is needed to improve readability.

Minor and specific comments are described below:

Abstract - It would be helpful to state in the abstract what the intended product from this alga is, and what characteristics are necessary (e.g. high carbohydrate content). This would help the reader evaluate whether the summarized results (e.g., increased carbohydrates and decreased lipids under mixotrophic conditions) are advantageous or disadvantageous for the intended product. Since the authors found that protein and antioxidants decreased under mixotrophic conditions, perhaps there should also be a statement about what specific types of products the algae can be used for after growing with cheese whey (i.e., products that just need carbohydrates).

Line 48: This sentence could be improved for clarity. Does the description “relative to the culture” mean “relative to the autotrophic culture”?

Line 73: Please clarify what “light invasion” means. Does this mean photoinhibition?

Line 102: It may be helpful to specify in the Introduction which biochemical components are advantageous for which algae products. For example, which products would need high carbohydrates and antioxidants, and which products would need high lipids. Please also define phenolic compounds and what product they are advantageous for.

Line 119: Please state if the culture was axenic (did not contain bacteria) or not.

Line 122: Please specify what type of percentage this means (e.g., volume per volume or mass per volume). Were nutrient concentrations the same across treatments and the control, even if different percentages of cheese whey were added?

Line 161: Does the added cheese whey affect optical density of the medium? Please explain how you used different standard curves because of this fact, and that they are available in the Supporting Information. (I believe Figure S2 is for the 2.5% whey treatment, but the caption says 5% whey).

Line 169: Here and in the Results section, the specific growth rate is often referred to as the maximum specific growth rate, which could be incorrect. Based on the description in the methods section, the authors are calculating the specific growth rate based on data measured in the experiment, not the maximum specific growth rate, which would require measuring many specific growth rates at different substrate concentrations.

Line 172: Writing out these equations for specific growth rate seems unnecessary, the authors could simply that the specific growth rate was calculated as the slope of the natural log of biomass concentration versus time during exponential phase, when the correlation coefficients of these two variables was above 0.98. Please also state which days were used to calculate the specific growth rate, since based on Figure 1 it does not look like the natural log of biomass versus time would be linear for the entire duration of the experiment.

Line 183-184: The description of the polynomial function seems unclear. Did the authors just calculate the productivity for each day compared to day zero using Eq. 2, or did they use another function to compare productivity between all different days?

Line 191: Could the authors briefly state how total nitrogen was measured?

Line 225: Please explain why two antioxidant methods were used. Was it to cross-check the results?

Table 1: How were these variables measured for the whey? Should they be included in the methods section?

Table 2: As stated previously, this variable is the specific growth rate, not the maximum specific growth rate, based on the calculation method described in the methods section. This table should also include standard deviations based on the triplicates.

Line 318: Authors are referring to a figure of productivity, so should clarify that they mean productivity and not specific growth rate.

Figure 4: Please include a y-axis label.

Table 4: Please explain these numbers, are they the lowest and highest fatty acid content (% of total fat?) of the three replicate cultures?

Please discuss the implications of having a higher ash content and lower antioxidant activity when whey is added to the medium at optimal percentages (5%). Does this change which industries would be able to use the algae?

7. PLOS authors have the option to publish the peer review history of their article (what does this mean?). If published, this will include your full peer review and any attached files.

Reviewer #1: No

Reviewer #2: Yes: Sarah E. Loftus

---

## [Author Response · Author response to Decision Letter 1]

13 Sep 2019

Dear Reviewers,

Response to reviewers

We would like to thank the reviewers for careful and thorough reading of this manuscript and for the thoughtful comments and constructive suggestions, which help to improve the quality of this manuscript. Our response follows (the reviewer’s comments are in italics and blue color). 

Reviewer #1: 

Highlight: Not provided

Graphical abstract: Not provided

General remarks: This manuscript focus on the growth of S. platensis microalgae using a pollutant by-product called cheese whey for high biomass and carbohydrate production. The used of English in this manuscript is poor and require to further amend the language. There is a lot of work needed to further improve this manuscript (e.g., structure arrangement, language, unnecessary introduction) My recommendation is not to accept this paper for now as there is still lack of assurance and required amendment in this manuscript. I hope my following comments world improve the following manuscript.

We agree with the reviewer that sufficient care was not taken in the original manuscript vis-à-vis the English and interpretations. In the revised manuscript, the English and grammatical errors are corrected and the interpretations and flow have been significantly improved. The translation of the manuscript to English has been corrected and revised by ECB – English Consulting Brazil (Annex A).

COMMENTS:

1) The used of English and grammar were poor. This section is not constructive enough in this section:

- Page 3, Line 63 – 65. Please read and cite some suggested paper for these biomolecules.

1. Khoo, K. S., Lee, S. Y., Ooi, C. W., Fu, X., Miao, X., Ling, T. C., & Show, P. L. (2019). Recent advances in biorefinery of astaxanthin from Haematococcus pluvialis. Bioresource technology, 121606.

2. Ren, H. Y., Xiao, R. N., Kong, F., Zhao, L., Xing, D., Ma, J., & Liu, B. F. (2019). Enhanced biomass and lipid accumulation of mixotrophic microalgae by using low-strength ultrasonic stimulation. Bioresource technology, 272, 606-610.

Also, please remove “bioactive compounds”.

Response: The entire paragraph has rewritten as per the reviewer’s suggestion including Line 63-65 (page 3) and these references have been added (current references 8 & 9) and briefly discussed in the text on p. 12

We have removed “Bioactive compounds”.

2) - Page 4. Line 90 – 104, please combine all into one paragraph.

Response: The entire section is now revised. The text has rearranged as per the reviewer’s suggestion as follows:

“Girard et al. [15] cultured Scenedesmus obliquus under mixotrophic conditions and demonstrated that this species presented higher specific growth rates and biomass yields in replacing 40% (v/v) of the standard ‘Bold’s basal medium’ (BBM) by whey. Salla et al. [2] studied the mixotrophic cultivation of S.platensis cyanobacterium in diluted Zarrouk growth medium with the addition of concentrated whey protein residues and high lactose levels. They observed that there was an increase in biomass and carbohydrate productivity by the species under study. Tsolcha et al. [16] developed a biological (algal) second cheese whey wastewater treatment system to generate renewable energy in the form of biodiesel, while simultaneously removing polluting nutrients and chemical oxygen demand. Recent studies demonstrate that S.platensis is also a promising bioethanol producer due to carbohydrate concentration in its biomass may reach 50% [2] and up to 60% [10] by changing the growth medium. In addition, the species may contain high content of antioxidant compounds being capable to add value to the process.”

3) Why is such concentration (2.5%, 5.0% and 10%) of cheese whey deployed for the culture medium? Why not (2.5%, 5.0% 7.5% and 10%?). Please give clarification for this. By doubling up the concentration is difficult for optimization.

Response: We appreciate the suggested from the reviewer. We agree with the reviewer that by doubling up the concentration is difficult for optimization in this study case and the optimization process has been one of the hot topics for the future of work since it was not the focus of this preliminary work. To examine the effect of whey addition on growth medium of S.platensis, the selected microalgae were triplicate cultured in mixotrophic conditions with 2.5%, 5.0% and 10.0% of cheese whey. The experimental logic for defining whey addition concentrations in the microalgae cultures was based on previous studies involving microalgae mixrotrophic cultivation (Salla et al. 2017; Mouther (2010)). Mouther (2010) showed that optimal cheese whey concentration for S. platensis mixotrophic cultivation was 3.0% and that concentration above 6% caused growth inhibition. Salla et al. 2017 evaluated S.platensis mixotrophic cultivation with residues obtained through the processes of ultra- and nanofiltration of whey at concentrations of zero, 1.25% and 2.5% (v/v). In this sense, it was observed that they evaluated the S.platenis mixotrophic cultivation by doubling up the concentration of the organic carbon source. Gao et al. 2019 examined the effect of initial TOC/TN (ratio of organic carbon to nitrogen) ratio of wastewater on the cultivation of microalgae, the selected microalgae were triplicate cultured in simulated wastewater with TOC/TN ratio of 0, 1, 3, 6, 12, 24 and 30, respectively.

References:

 Salla ACV, Margarites AC, Seibel FI, Holz LC, Brião VB, Bertolin TE, et al. Increase in the carbohydrate content of the microalgae Spirulina in culture by nutrient starvation and the addition of residues of whey protein concentrate. Bioresour Technol. 2016;209: 133-141. 

 Mourthé K. Obtenção de biomassa de Arthrospira platensis (Spirulina platensis) utilizando do soro de leite. Tese de doutorado, Universidade Federal de Minas Gerais. 2010. Available from: http://www.bibliotecadigital.ufmg.br/dspace/handle/1843/BUOS-8EJQW3.

 Feng Gao, Hong-Li Yang, Chen Li, Yuan-Yuan Peng, Yuan-Ming Guo. Effect of organic carbon to nitrogen ratio in wastewater on growth, nutrient uptake and lipid accumulation of a mixotrophic microalgae Chlorella sp. Bioresource Technology. 2019; 282:118-124.

4) Page 9, Line 266 – 269, the English used is too poor, please rephrase.

Response: The text (pag 9, line 266-269) has rewritten as per the reviewer’s suggestion as follows:

“The whey concentrations were based on previous studies: Mouther (2010) [26] showed that optimal cheese whey concentration for S. platensis mixotrophic cultivation was 3.0% and inhibition occurs at higher whey concentrations. Moreover, heterotrophic cultivation was inappropriate for S. platensis, possibly due to some inhibiting factor for this species present in the whey.”

5) Based on Figure 1, why did the author stop at Day 17 and state it’s the maximum growth? Why not increase to Day 20 to see if there is any growth?

Response: All cultures reached stationary phase after 17 days of cultivation. The microalgae were cultivated until the early stationary phase of growth, which is when carbohydrates are accumulated in the microalgal biomass according to Moura et al. 2006 and Salla et al. 2017. The stationary phase is often due to a growth-limiting factor such as the depletion of an essential nutrient, and/or the formation of an inhibitory product. Stationary phase results from a situation in which growth rate and death rate are equal. The number of new cells created is limited by the growth factor and as a result, the rate of cell growth matches the rate of cell death.

References: 

 Moura, A.M., Bezerra Neto, E., Koening, M.L., Leça, E.E.Chemical composition of microalgae in semi-intensive culture: Chaetoceros gracilis Schutt, Isochrysi sgalbana Parke and Thalassiosira weiss flogii (Grunow). Revista Ciência Agronômica. Volume 37, Issue 2, Pages 142-148

 .Salla ACV, Margarites AC, Seibel FI, Holz LC, Brião VB, Bertolin TE, et al. Increase in the carbohydrate content of the microalgae Spirulina in culture by nutrient starvation and the addition of residues of whey protein concentrate. Bioresour Technol. 2016;209: 133-141. 

6) There is a lot of new paragraph for each section. Advise author to keep 1-3 paragraph is sufficient enough.

Response: The entire manuscript is now revised. We have now completely rewritten this manuscript and have removed the superfluous and irrelevant discussions and related references as per the reviewer’s suggestion.

7) Page 10, Line 295 – 298, This paragraph does not make any sense and clueless on discussion. Please revise. 

Response: (Page 10, “line 295 – 298”) This paragraph has been deleted as it was considered to be superfluous.

8) I realize that the discussion part is too long and will get lost during reading. Advise author to revise overall of this manuscript. Keep it short and simple for the reasoning.

Response: The entire manuscript is now revised. We have now completely rewritten this manuscript and have removed the superfluous and irrelevant discussions and related references as per the reviewer’s suggestion.

9) Page 13, Line 376, what species? Please state it.

Response: The name of specie has been mentioned as follows:

“…the basic chemical composition of S. platensis consists of proteins (50 - 70%), carbohydrates (15 - 25%)…”

10) Page 17, Line 462 – 488. Combine all into one paragraph or remove it. It is too lengthy and not necessary for this result and discussion section. Much more to introduction.

Response: In Page 17, the text (“Line 462 – 488”) has been deleted as per reviewer’s suggestion and part of the text was moved from results and discussion to introduction. 

11) Please revise the structure of the overall manuscript. There are still too many new opening paragraphs which does not seem to be important.

Response: The entire manuscript is now revised. We have now completely rewritten this manuscript and have removed the superfluous and irrelevant discussions and related references as per the reviewer’s suggestion.

12) For Figure 2, why is the error bar for 10% whey cheese is so large. Some were small um erro experimental maior and some were large. This shows the inconsistency for the data collected.

Response: The monitoring of the growth of S. platensis cultures was performed by measuring the absorbance at λ = 670 nm in spectrophotometer (SP-22 Biospectro). A calibration curve was generated to relate the absorbance to the cell dry weight. The opacity of growth medium using cheese whey as organic carbon source increases with increasing whey concentration. The higher opacity of samples can make absorbance determination by spectrophotometry difficult increasing the experimental error but within acceptable limits. Moreover, the culture with 10% whey grew faster than the other cultures in the early days thus the high density of the cell is another factor that could increase experimental error by spectrophotometry requiring samples dilutions to maintain the linearity range of method. In Fig 2 appears that culture with 5% whey also showed a considerable standard deviation possibly for the same reasons. In this sense, is was expected that mixotrophic cultivation with 10% whey may lead to large experimental errors. Curves of dry weight biomass determined from optical density measurements from relevant study (Poddara et al. 2018) show that the standard deviation obtained from date of heterotrophic and mixotrophic growth were higher than those obtained with autotrophic growth.

Reference: 

1.Nature Poddara, Ramkrishna Senb, Gregory J.O. Martin. Glycerol and nitrate utilisation by marine microalgae Nannochloropsis salina and Chlorella sp. and associated bacteria during mixotrophic and heterotrophic growth, Algal Research 33 (2018) 298–309.

Reviewer #2: 

The authors used an experimental approach and determined the biochemical composition of the algae, in addition to biomass yields, specific growth rates, and antioxidant activity. The authors compared their results to those found in other studies, providing plenty of context. They found that Spirulina platensis can reach higher biomass yields and specific growth rates when clarified cheese whey is added to the growth medium, and that carbohydrate content increases. Protein, lipid, and antioxidant contents decreased, however, which affects which products can be made from the algae. Overall the cheese whey would be effective for improving biomass production and reducing cultivation costs for algae products that need high carbohydrate costs.

One major way the study could be improved is by performing statistical analyses, which are not described in the methods section. However, results throughout the abstract, results, and discussion sections are often described as significant, even though it appears statistical tests were not performed. If statistical analyses are not going to be performed, the authors should be transparent about this, and only use qualitative descriptors to present the results. It appears that all data underlying results of the study are not available in the supplementary information (e.g., data for each of the control and treatment replicates that are used to provide the average and standard deviations in tables and figures). The authors should further clarify how their current manuscript complies with the journal’s data availability policy, or should include more of the experimental data in supporting documents or ideally in an online public data repository (e.g., Figshare). Lastly, the intended meaning of the manuscript writing is mostly understandable, but further editing is needed to improve readability.

We agree with the reviewer that the study could be improved is by performing statistical analyses and it has been included in the work. All data underlying results of the study are now available in an online public data repository (Figshare)(DOI:10.6084/m9.figshare.9820181) and also in the supplementary information . In the revised manuscript, the English and grammatical errors are corrected and the interpretations and flow have been significantly improved. The translation of the manuscript to English has been corrected and revised by ECB – English Consulting Brazil (Annex A).

Minor and specific comments are described below:

1) Abstract - It would be helpful to state in the abstract what the intended product from this alga is, and what characteristics are necessary (e.g. high carbohydrate content). This would help the reader evaluate whether the summarized results (e.g., increased carbohydrates and decreased lipids under mixotrophic conditions) are advantageous or disadvantageous for the intended product. Since the authors found that protein and antioxidants decreased under mixotrophic conditions, perhaps there should also be a statement about what specific types of products the algae can be used for after growing with cheese whey (i.e., products that just need carbohydrates).

Response: Necessary correction has been incorporated as per reviewer’s suggestion as follows:

“Abstract: Mixotrophic cultivation of microalgae provides a very promising alternative for producing carbohydrate-rich biomass to convert into bioethanol and chemicals. It has been demonstrated that S. platensis may present high yields of biomass and carbohydrates when it is grown under mixotrophic conditions using cheese whey. However, there are no previous studies of this species to evaluate the influence of this culture system on the profile of fatty acids and the production of antioxidant compounds being extremely important for food and pharmaceutical applications as adding value to the process. S. platensis presented higher specific growth rates, biomass productivity and carbohydrate content under mixotrophic conditions, however the antioxidant capacity, protein and lipid content were lower than that of the autotrophic culture. The maximum biomass yield was 2.98 ±0.07 g/L in the growth medium with 5.0% whey. The phenolic compound concentration was the same for the biomass obtained in the autotrophic and mixotrophic conditions with 2.5% and 5.0% whey. The phenolic compound concentration no showed significant differences except in the growth medium with 10.0 % whey, which presented average values of 22.37±0.14 mg gallic acid/g. Mixotrophic cultivation of S. platensis using whey can be considered a viable alternative to reduce the costs of producing S. platensis biomass, shorten cultivation time and produce carbohydrates, as it does not require adding expensive substrates to the growth medium, while also taking advantage of cheese whey considered as a pollutant. Cultivation for S.platensis biomass production is the most widespread due to high biomass production capacity and it might be a promising bioethanol producer”.

2) Line 48: This sentence could be improved for clarity. Does the description “relative to the culture” mean “relative to the autotrophic culture”?

Response: This sentence has rewritten and the description “relative to the culture” was replaced by “relative to the autotrophic culture”.

3) Line 73: Please clarify what “light invasion” means. Does this mean photoinhibition?

Response: “Light invasion” has been deleted as per reviewer’s suggestion.

4) Line 102: It may be helpful to specify in the Introduction which biochemical components are advantageous for which algae products. For example, which products would need high carbohydrates and antioxidants, and which products would need high lipids. Please also define phenolic compounds and what product they are advantageous for.

Response: Thank you for your comment. This is a fair point. We have now included a new paragraph in the Introduction section, which addresses these issues.

4) Line 119: Please state if the culture was axenic (did not contain bacteria) or not.

Response: The culture was xenic and this information has been added in Material and methods section as per reviewer’s suggestion (Page X, Line 2-3) as follows:

“A xenic strain of Spirulina platensis was maintained under sterile conditions in test tubes with a temperature of 25°C and irradiance of 238 μmol.”

5) Line 122: Please specify what type of percentage this means (e.g., volume per volume or mass per volume). Were nutrient concentrations the same across treatments and the control, even if different percentages of cheese whey were added?

Response: Necessary correction has been incorporated as per reviewer’s suggestion as follows:

“…. For the mixotrophic cultures, the complete Zarrouk medium was supplemented with 2.5%(v/v), 5.0%(v/v) and 10%(v/v) buffalo mozzarella cheese whey….”

6) Line 161: Does the added cheese whey affect optical density of the medium? Please explain how you used different standard curves because of this fact, and that they are available in the Supporting Information. (I believe Figure S2 is for the 2.5% whey treatment, but the caption says 5% whey).

Response 1: The monitoring of the growth of S. platensis cultures was performed by measuring the absorbance at λ = 670 nm in spectrophotometer (SP-22 Biospectro). A calibration curve was generated to relate the absorbance to the cell dry weight. The opacity of growth medium using cheese whey as organic carbon source increases with increasing whey concentration. The higher opacity of samples may make absorbance determination by spectrophotometry difficult increasing the experimental error but within acceptable limits. In this sense, different standard curves were used for all cultures with a specific blank for each composition of growth medium. 

Response 2: Figure S2 is for the 2.5% whey treatment and the caption has been corrected.

7) Line 169: Here and in the Results section, the specific growth rate is often referred to as the maximum specific growth rate, which could be incorrect. Based on the description in the methods section, the authors are calculating the specific growth rate based on data measured in the experiment, not the maximum specific growth rate, which would require measuring many specific growth rates at different substrate concentrations.

Response: We replaced the term (the maximum specific growth) for the correct one (the specific growth rate) as per reviewer’s suggestion.

8) Line 172: Writing out these equations for specific growth rate seems unnecessary, the authors could simply that the specific growth rate was calculated as the slope of the natural log of biomass concentration versus time during exponential phase, when the correlation coefficients of these two variables was above 0.98. Please also state which days were used to calculate the specific growth rate, since based on Figure 1 it does not look like the natural log of biomass versus time would be linear for the entire duration of the experiment.

Response 1: The equation for specific growth rate has been deleted and the topic has rewritten as follows:

“The growth rate (μ d-1) of S. platensis was calculated as the slope of the natural log of biomass concentration versus time during exponential growth phase, when the correlation coefficients of these two variables was above 0.98 [20].

Response 2: We agree that the natural log of biomass versus time was not linear for the entire duration of the experiment. The specific growth rates were calculated only in the exponential phase of growth. The duration time of exponential growth phase was different for each evaluated cultivation condition and it was considered in the calculations. 

9) Line 183-184: The description of the polynomial function seems unclear. Did the authors just calculate the productivity for each day compared to day zero using Eq. 2, or did they use another function to compare productivity between all different days?

Response: We just calculated the productivity for each day compared to day zero using Eq. 2. Necessary changes have been made and this part is now rewritten as follows:

“The maximum biomass productivity PXmax (g.L.dia-1), were evaluated according to Eq. (1), where Xmax is biomass maximum concentration at time t, Xo is the biomass concentration at time zero, and t is time. 

P_(Xmax=) (X_max-Xo)/t (1)

 The productivity was calculated for each day compared to day zero. The maximum value obtained was defined as the maximum biomass productivity.”

10) Line 191: Could the authors briefly state how total nitrogen was measured?

Response: We briefly stated how total nitrogen was measured in material and methods section as follows: 

“Kjeldahl nitrogen was measured (TECNAL digester block, model TE-040; TECNAL Kjeldahl TE-0364 distillation unit) using 200 mg samples of the dry biomass. The sample was first digested using concentrated sulfuric acid digester solution following a progressive heating ramp, and then distilled using boric, sodium hydroxide and indicator solutions. After distillation, the obtained solution was titrated with 0.1 N chloridic acid (HCL). The total protein content was calculated by multiplying the value for total nitrogen by the conversion factor of 4.78 [21].”

11) Line 225: Please explain why two antioxidant methods were used. Was it to cross-check the results?

Response: Antioxidants are compounds capable to either delay or inhibit the oxidation processes which occur under the influence of atmospheric oxygen or reactive oxygen species. They are used for the stabilization of polymeric products, of petrochemicals, foodstuffs, cosmetics and pharmaceuticals. The various analytical methods of evaluation of the antioxidant capacity fall into distinct categories. Due to the different types of free radicals and their different forms of action in living organisms, there is no a simple and universal method to quantify antioxidant activity. In this sense, different antioxidant methods were tested in this work. The ABTS method: The ABTS cation radical (ABTS•+) is formed by the loss of an electron by the nitrogen atom of ABTS (2,2’-azino-bis(3- ethylbenzthiazoline-6-sulphonic acid)). In the presence of Trolox (or of another hydrogen donating antioxidant), the nitrogen atom quenches the hydrogen atom, yielding the solution decolorization. The FRAP (ferric reducing antioxidant power) method: The FRAP (ferric reducing antioxidant power) method relies on the reduction by the antioxidants, of the complex ferric ion-TPTZ (2,4,6-tri(2-pyridyl)- 1,3,5-triazine). The binding of Fe2+ to the ligand creates a very intense navy blue color. 

These references show different methods to evaluate antioxidant activity of biomass:

References: 

 Zhendong Yang, Weiwei Zhai. Identification and antioxidant activity of anthocyanins extracted from the seed and cob of purple corn (Zea maysL.) Innovative Food Science and Emerging Technologies 11 (2010) 169–176.

2. Sonia Milla, Edgar Uquiche. Antioxidant activity of supercritical extracts from Nannochloro psisgaditana: Correlation with its content of carotenoid sand to copherols. J.of Supercritical Fluids 111 (2016) 143–150.

12) Table 1: How were these variables measured for the whey? Should they be included in the methods section?

Response: These methods have been included in the in the methods section

“Clarified buffalo cheese whey samples were analyzed for percentage of total solids, total protein, fat, lactose, and moisture using the DairySpec (Bentley Instruments Inc, Chska Minnesota, USA). The equipment was calibrated using buffalo cheese whey calibration sample with different concentration ranges.”

13) Table 2: As stated previously, this variable is the specific growth rate, not the maximum specific growth rate, based on the calculation method described in the methods section. This table should also include standard deviations based on the triplicates.

Response: We replaced the term (the maximum specific growth) for the correct one (the specific growth rate). We included standard deviations based on the triplicates (Xmax(g.L-1), Pmax(g.L-1.day-1), µ (day-1)) in Table 2 as per reviewer’s suggestion.

14) Line 318: Authors are referring to a figure of productivity, so should clarify that they mean productivity and not specific growth rate.

Response: Necessary changes have been made and this part is now rewritten as follow:

“The culture with 10.0% whey showed double the specific growth rate of the culture with 5.0% and it had the highest biomass productivity on the 4th day (Fig 2).”

15) Figure 4: Please include a y-axis label.

Response: Necessary changes have been made and y-axis label is included in Figure 4.

16) Table 4: Please explain these numbers, are they the lowest and highest fatty acid content (% of total fat?) of the three replicate cultures?

Response: A gas chromatograph (Thermo Scientific-CG/FID-FOCUS) with flame ionization detector (FID) and Supelco SP2560 capillary column (100m x 0.25mm x 0.2μm) was used to determine the fatty acid profile. This method is accurate but it is costly so we chose to do these analyzes in duplicate. These numbers are the lowest and highest (FAMEs) content.” (Percentage of total FAMEs).

This section has been rewritten as follows:

‘’ The fatty acid methyl esters (FAMEs) profiles of S. platensis are shown in Table 4. 

Table 4 - The fatty acid methyl esters (FAMEs) profile (percentage of total FAMEs) present in S. platensis biomass cultured in control medium and mixotrophic medium with the addition of buffalo cheese whey (These numbers are the lowest and highest (FAMEs) content).”

17) Please discuss the implications of having a higher ash content and lower antioxidant activity when whey is added to the medium at optimal percentages (5%). Does this change which industries would be able to use the algae?

Response: The maximum cell concentration (2.98 g/L) was obtained in experiment with addition of %5 whey. There has been a 70% and 76% increase in biomass production and carbohydrates content, respectively. With this culture conditions, the increase in carbohydrate production in S.platensis indicated this specie as potential biomass for production of bioethanol. These data would help to explain the microalgal growth and chemical compositions, and enhance the knowhow on improving the performance of mixotrophic cultivation. Moreover, the results of present study support that mixotrophic cultivation of S.platensis with 5% whey could be a viable strategy to the production of phenolic compounds and/or antioxidant compounds by lower cost optimized process since the biomass productivity was higher in this condition.

S. platensis is a species that has been studied in thermal degradation processes for producing bio-oil, biochar and gas. Some studies have shown that bio-oil produced from microalgae are more stable, have lower oxygen content and higher calorific value than lignocellulosic bio-oil. However, bio-oil presents high nitrogen content that is undesirable because it deactivates acid catalysts used for co-processing crude oil in refineries and emits NOx during combustion. The mixotrophic culture of S. platensis using cheese whey in addition to increasing biomass production may decrease the processing cost and produce low protein biomass, thus being favorable to applications in thermal degradation processes for producing biofuels. Huang et al. 2016 evaluated the bio-oil production from hydrothermal liquefaction (HTL) of high-ash microalgae. This work focused on high-ash microalgae, a representative of algae with practical importance which were widely cultivated in wastewater or other nitrogen unlimited environment. The microalgae feedstocks had relatively high ash contents (28.92 wt%, 35.23 wt%). This study demonstrated the feasibility of applying HTL to produce bio-oil from high-ash microalgae, and the findings on bio-oil properties and transfer behavior of carbon and nitrogen supplied useful information for downstream utilization. Liu et al. 2019 evaluated the hydrothermal carbonization (HTC) of natural microalgae containing a high ash content. The char generated from HTC is called as hydrochar. The hydrochar derived from microalgae has large irregular porosity aggregates and higher cation exchange capacity, which is different from the lignocellulose biochar. The higher nutrient content of nitrogen, ash and inorganic elements are beneficial in agriculture. Furthermore, hydrochar containing high aromaticity structures has the potential to be solid fuel. Many researches show that mineral salt as catalysts have been proven to play important roles in the yield and quality of products. The results of this study revealed that natural microalgae can be utilized by hydrothermal carbonization to generate renewable fuel resources.

References:

 Yanqin Huang, Yupeng Chen, Jianjun Xie, Huacai Liu, Chuangzhi Wu. Bio-oil production from hydrothermal liquefaction of high-protein high-ash microalgae including wild Cyanobacteria sp. And cultivated Bacillariophyta sp. Fuel. 2016;183:9-19.

 Huihui Liu, Yingquan Chen, Haiping Yang, Francesco G. Gentili, Hanping Chen. Hydrothermal carbonization of natural microalgae containing a high ash content. Fuel. 2019;249: 441-448.

---

## [Editor Report · Decision Letter 2]

19 Sep 2019

PONE-D-19-15520R2

Mixotrophic cultivation of Spirulina platensis in dairy wastewater: effect on the production of biomass, biochemical composition and antioxidant capacity

PLOS ONE

Dear Dr. Chagas,

Thank you for submitting your manuscript to PLOS ONE. After careful consideration, we feel that it has merit but does not fully meet PLOS ONE’s publication criteria as it currently stands. Therefore, we invite you to submit a revised version of the manuscript that addresses the points raised during the review process.

Alhough they addressed most of the issues perceived by reviewers, authors still failed to present their manuscript in a scientifically acceptable form. The text lacks readability and abounds superficial espressions. Moreover, syntax is generally poor. In the presented form the manuscript cannot be accepted for publication. I strongly encourage authors to ask for a critical review of a senior colleague(s) to give frank and objective advices how to reformat the text to gain a publishable shape. If authors wish, I can provide examples of my concerns in future correspondence.

We would appreciate receiving your revised manuscript by Nov 03 2019 11:59PM. To enhance the reproducibility of your results, we recommend that if applicable you deposit your laboratory protocols in protocols.io, where a protocol can be assigned its own identifier (DOI) such that it can be cited independently in the future. For instructions see: http://journals.plos.org/plosone/s/submission-guidelines#loc-laboratory-protocols

We look forward to receiving your revised manuscript.

Kind regards,

Branislav T. Šiler, Ph.D.

Academic Editor

PLOS ONE

---

## [Author Response · Author response to Decision Letter 2]

8 Oct 2019

Response to Editor

Editor’s Comment: 1..Please upload a copy of Figures 1,2,3 which you refer to in your text . Or if the figure is no longer to be included as part of the submission please remove all reference to it within the text.

The copy of Figures 1, 2 and 3 have been uploaded.

Best,

Bruna Chagas

---

## [Editor Report · Decision Letter 3]

10 Oct 2019

Mixotrophic cultivation of Spirulina platensis in dairy wastewater: effect on the production of biomass, biochemical composition and antioxidant capacity

PONE-D-19-15520R3

Dear Dr. Chagas,

We are pleased to inform you that your manuscript has been judged scientifically suitable for publication and will be formally accepted for publication once it complies with all outstanding technical requirements.

With kind regards,

Branislav T. Šiler, Ph.D.

Academic Editor

PLOS ONE
---

## [Editor Report · Acceptance letter]

17 Oct 2019

PONE-D-19-15520R3 

Mixotrophic cultivation of *Spirulina platensis* in dairy wastewater: effects on the production of biomass, biochemical composition and antioxidant capacity 

Dear Dr. Chagas:

I am pleased to inform you that your manuscript has been deemed suitable for publication in PLOS ONE. Congratulations! Your manuscript is now with our production department. 

With kind regards,

on behalf of

Dr. Branislav T. Šiler 

Academic Editor

PLOS ONE